# Component Decomposition-Based Hyperspectral Resolution Enhancement for Mineral Mapping

**Puhong Duan** [1,2], **Jibao Lai** [3], **Pedram Ghamisi** [2], **Xudong Kang** [1,*], **Robert Jackisch** [2], **Jian Kang** [4] **and Richard Gloaguen** [2]

1   College of Electrical and Information Engineering, Hunan University, Changsha 418002, China; puhong_duan@hnu.edu.cn
2   Helmholtz-Zentrum Dresden-Rossendorf (HZDR), Helmholtz Institute Freiberg for Resource Technology, 09599 Freiberg, Germany; p.ghamisi@hzdr.de (P.G.); r.jackisch@hzdr.de (R.J.); r.gloaguen@hzdr.de (R.G.)
3   Earth Observation System and Data Center, China National Space Administration, Bejing 100048, China; laijibao@163.com
4   Faculty of Electrical Engineering and Computer Science, Technical University of Berlin, 10587 Berlin, Germany; jian.kang@tu-berlin.de
*   Correspondence: xudong_kang@hnu.edu.cn; Tel.: +86-731-8882-2866

**Abstract:** Combining both spectral and spatial information with enhanced resolution provides not only elaborated qualitative information on surfacing mineralogy but also mineral interactions of abundance, mixture, and structure. This enhancement in the resolutions helps geomineralogic features such as small intrusions and mineralization become detectable. In this paper, we investigate the potential of the resolution enhancement of hyperspectral images (HSIs) with the guidance of RGB images for mineral mapping. In more detail, a novel resolution enhancement method is proposed based on component decomposition. Inspired by the principle of the intrinsic image decomposition (IID) model, the HSI is viewed as the combination of a reflectance component and an illumination component. Based on this idea, the proposed method is comprised of several steps. First, the RGB image is transformed into the luminance component, blue-difference and red-difference chroma components (YCbCr), and the luminance channel is considered as the illumination component of the HSI with an ideal high spatial resolution. Then, the reflectance component of the ideal HSI is estimated with the downsampled HSI image and the downsampled luminance channel. Finally, the HSI with high resolution can be reconstructed by utilizing the obtained illumination and the reflectance components. Experimental results verify that the fused results can successfully achieve mineral mapping, producing better results qualitatively and quantitatively over single sensor data.

**Keywords:** hyperspectral image; mineral mapping; resolution enhancement; intrinsic image decomposition

## 1. Introduction

Hyperspectral scanners, as a newly appearing technique in the mining field, have been extensively utilized to explore minerals, since hyperspectral images (HSIs) are able to record rich spectral information varying from visible to infrared wavelength in hundreds of spectral channels [1–9]. This characterization makes HSI record the reflectance spectrum profiles of different minerals, which allows a non-destructive and non-invasive way for the exploration of mineral deposits [10–12]. The main goal of mineral mapping is to decide the spatial location of various minerals. A fusion of spectral and spatial information with increased resolution provides not only enhanced qualitative information on surface mineralogy, but also specific material interactions of composition and structure. Geologists are able to map formerly undetectable geological features to extract structural and

mineralogical properties. Intrusions and mineralization found in dykes and veins or structures tied to tectonic forces, related to faults and folds, can be measured. However, current hyperspectral sensors cannot capture data with high resolution in terms of both spatial and spectral dimensions because of the finite sun irradiance. Therefore, the captured data often suffer from low spatial resolution, which causes limitations in identifying different minerals [13]. Different from HSIs, RGB images usually provide higher spatial resolution but much lower spectral resolution, i.e., R, G, and B three channels. Thus, the fusion of hyperspectral and RGB images is an effective scheme to yield a higher spatial and spectral resolution data, which is helpful for mapping all kinds of minerals.

In order to better achieve mineral mapping, several data fusion techniques have been developed in decades [10,14,15]. In [14], a decision-level multi-sensor fusion method was proposed based on RGB and three types of infrared data from short-wave to long-wave infrared HSIs. The low-rank component analysis was applied to extract the discriminative features of multi-sensor data, and then, a majority voting rule is used to select the final probability map. In [10], a fusion framework of VNIR and SWIR data was proposed based on majority voting rule, in which an automatic high-resolution mineralogical imaging system was used to generate training labels. These approaches to mineral mapping mainly focus on the application of existing machine learning algorithms.

In recent years, various super-resolution schemes of HSIs [16–22] have been designed to enhance the resolution, which can be loosely divided into two classes: (1) Fusion of hyperspectral and panchromatic data and (2) fusion of multispectral (MS) and hyperspectral data. The goal of hyperspectral and panchromatic image (PAN) fusion is to merge the spatial details of the PAN into each band of the HSI, such as component substitution (CS)-based schemes [23,24], multiresolution analysis (MRA)-based methods [25,26], and deep learning approaches [27,28]. For example, in [29], a guided filtering method was applied for the fusion of HSI and PAN data. In [30], a hyperspectral pansharpening approach was proposed by using the homomorphic filtering and matrix decomposition. In [31], an HSI pansharpening approach based on deep priors was proposed to boost the spatial resolution with the help of a high-resolution panchromatic image.

Fusion of MS and HSI aims at merging the spatial resolution of a MS and the spectral information of an HSI together so as to produce a high spatial resolution HSI. A few examples of such approaches are matrix factorization [32,33] and deep learning [34,35]. For instance, in [36], a coupled nonnegative matrix factorization method was proposed to merge multispectral and hyperspectral data via unsupervised unmixing. In [37], a novel coupled sparse tensor factorization method was developed for the fusion of MS and HSI, in which the HSI was considered as three modes and a sparse core tensor. In [38], a deep convolutional neural network (CNN) with a two-stream framework was developed to merge HSI and MS, in which the CNN was utilized to extract the deep features of the input data followed by fully connected layers.

In this work, we propose an effective approach to enhance the spatial resolution of HSIs with the guidance of RGB image for mapping minerals, since RGB images are easily obtained in practical applications and have a higher spatial resolution with respect to other modalities. In the proposed method, the ideal high resolution HSI (HR-HSI) is considered as dot multiplication of two components, i.e., illumination and reflectance components, based on the principle of IID (see Figure 1), which contains the following steps. First, the RGB image is transformed into the luminance component, blue-difference and red-difference chroma components (YCbCr) so as to yield the spatial details of the original RGB image, and the luminance channel is considered as an approximation of the illumination component of HR-HSI. Then, the reflectance component of the HR-HSI is estimated by combining the downsampled HSI and the downsampled luminance channel. Finally, the estimated illumination and reflectance components are reconstructed to obtain the HR-HSI.

- Inspired by the principle of IID, we propose a novel hyperspectral resolution enhancement method for mineral mapping via component decomposition. To our knowledge, this is the first time to formulate the resolution enhancement of HSI as an intrinsic decomposition model.

- The proposed approach makes the best use of the spatial details of RGB image and the rich spectral information of HSI to obtain the high resolution hyperspectral images. Moreover, the proposed method is more efficient and faster, which is quite suitable to be used in real applications.
- We investigate whether the spatially enhanced HSI obtained by fusing HSI and RGB data can preserve spectral fidelity and consequently be conducive to mineral mapping. Experimental results demonstrate that the fused results of HSI and RGB data produced by the proposed approach are beneficial for mapping minerals compared to other approaches.

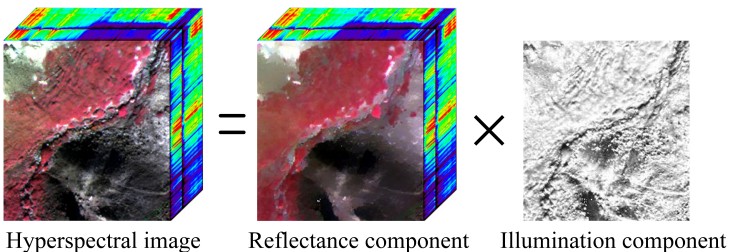

Hyperspectral image　　　Reflectance component　　　Illumination component

**Figure 1.** The principle of intrinsic image decomposition.

The remaining parts of this work are summarized as follows. Section 2 shows briefly intrinsic image decomposition. Section 3 is on the detailed steps of our method. The fusion results and discusses are exhibited in Section 4. Finally, conclusions are presented in Section 6.

## 2. Intrinsic Image Decomposition

IID is a challenging problem in image processing field [39,40], which is to divide an image into two components, i.e., the illumination component which is related to the bright property of the scene, and the reflectance component which reflects the material of different objects. The general model can be expressed as:

$$\mathbf{I} = \mathbf{R} \cdot \mathbf{S} \tag{1}$$

where $\mathbf{I}$ is the input. $\mathbf{R}$ and $\mathbf{S}$ are the illumination and reflectance components, respectively. It can be obseved from Equation (1) that estimating $\mathbf{R}$ and $\mathbf{S}$ with only $\mathbf{I}$ is an ill-posed issue. Currently, a variety of approaches have been developed to solve this issue by adding some prior information [41–43], which has been widely applied in image fusion, classification, and denoising. Different from those publications, in this paper, we only exploit the principle of the intrinsic image decomposition to perform the resolution enhancement of the HSI, which can greatly increase the computational efficiency. Therefore, the main goal of the proposed method focuses on estimating the reflectance and illumination components by utilizing the RGB and low-resolution HSI data.

## 3. Proposed Method

To obtain the HR-HSI for mineral mapping, we propose a component decomposition-based resolution enhancement method. Figure 2 presents the schematic of the proposed approach, which mainly contains three steps: First, the RGB image is transformed into the YCbCr space so as to obtain an estimation of the illumination component. Second, the reflectance component of the HR-HSI is calculated via using the downsampled HSI and the downsampled illumination component. Finally, the estimated illumination and reflectance components are combined together to reconstruct the HR-HSI.

As described before, the HR-HSI $\mathbf{I}_F$ is modeled as dot multiplication of the illumination component $\mathbf{S}_H$ and reflectance component $\mathbf{R}_H$, expressed as:

$$\mathbf{I}_F = \mathbf{S}_H \cdot \mathbf{R}_H \tag{2}$$

Equation (2) is an ill-posed inverse issue. The solution can be obtained by adding some priors. In this paper, instead of solving the complicated optimization problem, our goal is to calculate the illumination component $\mathbf{S}_H$ and reflectance component $\mathbf{R}_H$ by using the LR-HSI and RGB data, which makes our method more efficient compared to objective optimization methods.

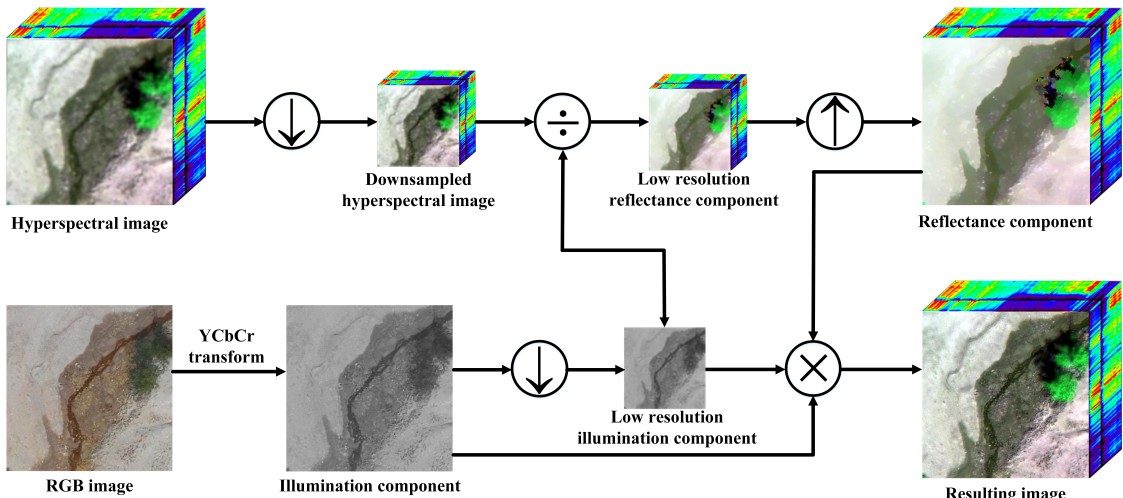

**Figure 2.** The flow chart of the proposed approach.

### 3.1. Estimation of the Illumination Component

The illumination component obtained by the intrinsic image decomposition mainly reflects the spatial details of the input. To fully merge the spatial details of the RGB image into HSI, the RGB image $\mathbf{I}_V$ is converted into the YCbCr space [44]. As shown in Figure 3, it can be observed that the luminance channel in the YCbCr space mainly records the spatial details of the RGB image, while the chrominance channels reflect the spectral information of the RGB image. Based on this characteristic, the luminance channel in the YCbCr space is considered as an estimation of the illumination component.

$$
\begin{cases}
\mathbf{Y} = 0.257 * \mathbf{R} + 0.564 * \mathbf{G} + 0.098 * \mathbf{B} + 16 \\
\mathbf{Cb} = -0.148 * \mathbf{R} - 0.291 * \mathbf{G} + 0.439 * \mathbf{B} + 128 \\
\mathbf{Cr} = 0.439 * \mathbf{R} - 0.368 * \mathbf{G} - 0.071 * \mathbf{B} + 128
\end{cases}
\tag{3}
$$

where $\mathbf{Y}$ represents the estimated illumination component $\mathbf{S}_H$. $\mathbf{R}$, $\mathbf{G}$ and $\mathbf{B}$ indicate three bands of RGB image.

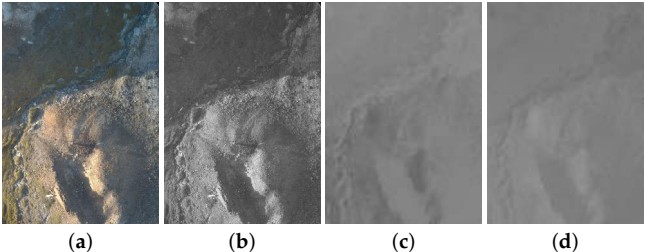

| (a) | (b) | (c) | (d) |

**Figure 3.** An example of luminance component, blue-difference and red-difference chroma components (YCbCr) transform on the RGB image. (**a**) RGB image. (**b**) Intensity channel. (**c**) Blue-difference chroma component. (**d**) Red-difference chroma component.

### 3.2. Estimation of the Reflectance Component

The reflectance component obtained by intrinsic image decomposition has relatively low spatial resolution. In addition, it mainly records the spectral information of the input. In this study, in order

to yield an accurate estimation of the reflectance component, the bicubic downsampling method is performed first on the original HSI so as to yield the corresponding low resolution HSI $\mathbf{I}_L$, where the downsampling scale is $1/4$. Then, the same downsampling operation is conducted on the $\mathbf{S}_H$ to obtain a low resolution illumination component $\mathbf{S}_L$. It should be mentioned that the main aim of the downsampling operation is to yield the low spatial data since the reflectance component is in low spatial resolution according to the principle of intrinsic image decomposition. Finally, the low resolution reflectance component $\mathbf{R}_L$ can be obtained as follows:

$$\mathbf{R}_L = \frac{\mathbf{I}_L}{\mathbf{S}_L} \tag{4}$$

*3.3. Reconstruction*

When the low resolution reflectance component $\mathbf{R}_L$ is obtained, the high resolution reflectance component $\mathbf{R}_H$ can be easily estimated with bicubic upsampling ($4\times$) on $\mathbf{R}_L$: $\mathbf{R}_H =\uparrow \mathbf{R}_L$. According to the principle of the intrinsic image decomposition (Equation (1)), the reconstructed HR-HSI can be produced by combining the high resolution reflectance component $\mathbf{R}_H$ and the estimated illumination component $\mathbf{S}_H$.

$$\mathbf{F} = \mathbf{R}_H \cdot \mathbf{S}_H \tag{5}$$

where $\mathbf{F}$ is the reconstructed high resolution HSI.

## 4. Experiments

In the experiment section, in order to verify the fusion performance of our approach, several advanced hyperspectral resolution enhancement approaches that have been achieved satisfactory performance in enhancing resolution of HSI, including CS methods, MAR approaches, and matrix factorization approaches. For CS methods, Gram-Schmidt (GS) [24] and principal component analysis (PCA) [45] are classic and effective resolution enhancement methods which have been widely used in some commercial softwares, e.g., the Environment for Visualizing Images (ENVI) and Earth Resources Data Analysis System (ERDAS). Therefore, they are considered as comparison approaches. For multi-resolution analysis methods, three representative approaches, i.e., smoothing filter-based intensity modulation (SFIM) [25], modulation transfer function-generalized Laplacian pyramid (MTF_GLP) [46], and high pass modulation (MGH) [47], are selected as comparison. For matrix factorization approaches, coupled nonnegative matrix factorization (CNMF) [36], hyperspectral subspace regularization (HySure) [48] are adopted for comparison since they can obtain a competitive fusion performance for resolution enhancement. For these approaches, the default parameters shown in the corresponding publications are used.

*4.1. Datasets*

(1) Disko dataset: The used HSI-RGB images were acquired during a geological remote sensing field campaign within the MULSEDRO project. The target area is the north shore of Disko Island in West Greenland ($69°885$ N, $52°577$ W). An intriguing geologic feature named Illukunnguaq dyke with a NW-SE strike direction is one target area. The lava intrusion of paleocene age is 5 m broad and passes through cretaceous sandstone-sediment formations. Illukunnguaq feature can be followed for roughly 800 m along the coastline and is known for iron-sulphide mineralization, containing Nickel and Copper.

Use of unmanned aerial vehicles (UAV) (Tholeg Octocopter Tho-R-PX-8/12) is a probate tool to acquire HSI and RGB imagery in high resolutions. A fixed-wing, the eBee+ UAV obtained high-resolution orthoimages form the target area and its surroundings at 20 MP per image with a SODA camera, where the captured image was geo-tagged using the drone's built-in GPS/GNSS receiver. Stereo-photogrammetry and Structure-from-motion via Agisoft Photoscan software created a detailed, georeferenced orthomosaic with 5 cm ground sampling distance (GSD) in RGB colour space.

HSI image mosaics were scanned with the Senop Rikola frame-based camera, having a resolution of 0.6 MP and 50 image channels in flight-mode. The Senop Rikola camera operates in the spectral range covering 504–900 nm, with a spectral resolution of 15 nm in average per band. Figure 4 shows the three band composite of HSI and RGB image. This scene contains six types of land covers, including vegetation, sandstone, basalt, sulphide, debris, and sandstone-basalt. Flying close to the surface target, the camera takes images on top of GPS points, which requires the UAV to complete a stop-scan-and-motion pattern along preprogrammed flight vectors. Pre-processing of HSI comprised of geometric and radiometric corrections which were accomplished in the MEPHySTo software tool [49]. The resulting HSI mosaic measures 350 × 50 m with a resampled GSD of 14 cm. The spatial size of the HSI is 1992 × 1531 pixels. External acquisition conditions for this dataset were favourable, having sunny illumination and weak winds for smooth UAV flights. Application of the described UAS workflow proved to create valuable geologic information in related arctic scenarios [50].

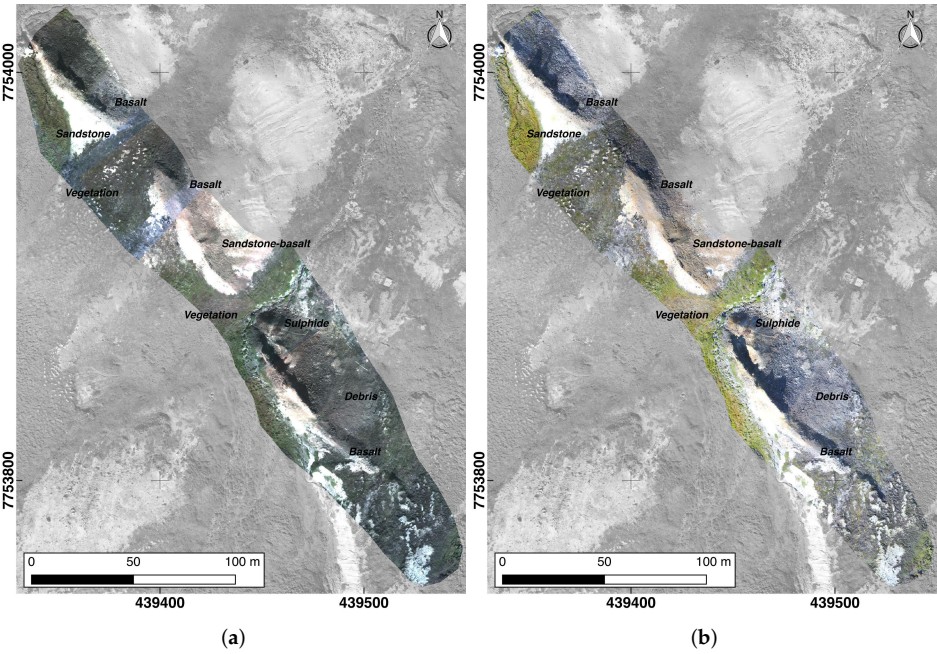

**Figure 4.** Disko dataset. (**a**) False color composite of hyperspectral image (HSI) (No. 17, 7, 1 at 61 nm, 551 nm, 504 nm). (**b**) RGB image.

(2) Litov dataset: This particular dataset was acquired with the very same method as for the first dataset. However, on the premises of a mine tailing area (50°158 N, 12°530 E) in the Sokolov region of the Czech Republic [51] in summer 2018. Residuals of lignite mining including soils and brown coal were dumped, sealed, and renatured with vegetation. Yet, the natural phenomenon referred to as acid mine drainage occurs along the SW border of the tailing. Acidic waters (pH 2–4) with increased loads of heavy metals, sulphur and reduced oxygen concentration drain from tailing channels towards an artificial lake. Precipitation of iron-bearing proxy minerals along the seams of said streams and their surroundings can be observed and detected by hyperspectral image analysis. Again, an enhanced resolution leads to an exact exposure of affected areas. The very same methods as in the Disko dataset were applied for documentation and preprocessing of the data. The GSD of the dataset is 2.5 cm for RGB and 3.7 cm for HSI, respectively. The captured HSI measures 20.5 m × 33.5 m for the Litov scene, and the spatial size of this image is 1066 × 909 pixels. Figure 5 shows the three band composite of HSI and RGB image. This scene shows a small canyon area with a stream, bordering the tailings.

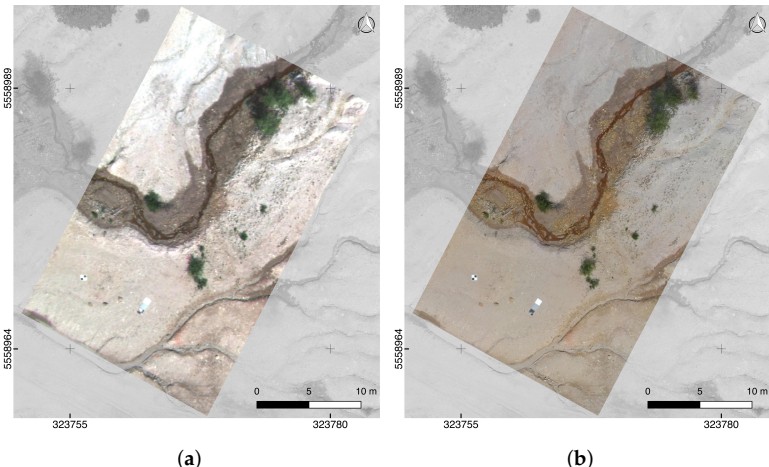

(**a**)                                                                                        (**b**)

**Figure 5.** Litov dataset. (**a**) False color composite of HSI (No. 17, 7, 1 at 61 nm, 551 nm, 504 nm). (**b**) RGB image.

### 4.2. Quality Indexes

In this work, we assess the resolution enhancement performance of different approaches in terms of both visual images and objective qualities. For the visual effect, we mainly observe the spatial details of different methods. For the objective evaluation, four mostly used quality indexes, which are described briefly, are adopted in our study, including cross correlation (CC) [52], spectral angle mapper (SAM) [16], root mean squared error (RMSE) [16], and erreur relative globale adimensionnelle de synthèse (ERGAS) [53]. The CC measures the spatial information. The SAM estimates the spectral similarity. The RMSE and ERGAS denote the global spatial and spectral information. All quality indexes are obtained by comparing the reconstructed HSI and the original HSI.

(1)    CC: The CC estimates the similar level of the original image and the resulting image:

$$CC(X, \hat{X}) = \frac{1}{N} \sum_{i=1}^{N} CCS(X_i, \hat{X}_i) \tag{6}$$

where

$$CCS(X, \hat{X}) = \frac{\sum\limits_{i=1}^{M} (X - \mu_X)(\hat{X} - \mu_{\hat{X}})}{\sqrt{\sum_{i=1}^{M} (X - \mu_X)^2 \sum_{i=1}^{M} (\hat{X} - \mu_{\hat{X}})^2}} \tag{7}$$

Here, $X$ is the reference image, and $\hat{X}$ denotes the fused image. A higher CC indicates the better fusion performance.

(2)    SAM: The SAM reflects the spectral quality of the reconstructed result, which is shown as:

$$SAM(X, \hat{X}) = \frac{1}{M} \sum_{i=1}^{M} ar\cos \frac{\hat{X}_i^T X_i}{\left\| \hat{X}_i \right\|_2 \left\| X_i \right\|_2} \tag{8}$$

The SAM is an important index of the spectral distortion of the fused result. A smaller SAM indicates less spectral distortion of the resulting image.

(3)    RMSE: The RMSE evaluates the difference between the fused result and the reference data, which is given as

$$RMSE(X, \hat{X}) = \frac{\sqrt{trace[(X - \hat{X})^T (X - \hat{X})]}}{\sqrt{N * M}} \tag{9}$$

The smaller value indicates better performance. The best value is 1.

(4)    ERGAS: The ERGAS assesses the overall quality of the fused result as follows:

$$ERGAS(X, \hat{X}) = \frac{100}{c} \sqrt{\frac{1}{M} \sum_{i=1}^{M} \frac{MSE(X_i, \hat{X}_i)}{\mu_{\hat{X}^i}^2}} \tag{10}$$

where $c$ represents the ratio of the spatial resolution between the fused result and the reference data. $\mu_{\hat{X}^i}^2$ denotes the mean value of $\hat{X}^i$. $MSE(X_i, \hat{X}_i)$ defines the mean square error between $X_i$ and $\hat{X}_i$. The smaller the ERGAS, the better the resulting image is.

### 4.3. Resolution Enhancement Results

In this subsection, the fused results of all approaches will be presented and discussed.

#### 4.3.1. Disko Dataset

Figure 6a,b presents the RGB image and the downsampled HSI (with scale 4), respectively. Figure 6c–j presents the results of resolution enhancement obtained by all methods. The HySure method produces obvious spectral distortion at the plant region. The GS, PCA, and CNMF methods also exhibit spectral distortion at the shadow region of the resulting images. The result obtained by the MLP_GLP approach looks blurred compared to the MGH method. The SFIM method slightly improves the fusion performance in enhancing the spatial details. In contrast, our method can restore more detailed information compared to other approaches (see the local enlarged region in Figure 6j). Furthermore, in order to further illustrate the spectral preservation ability, the spectral reflectance values of the raw HSI and different methods at two different locations are given in Figure 7. We can observe that the spectral reflectance of our method is closer to the reflectance of the raw HSI among all compared methods, which also illustrates that our method performs well in preserving the spectral information of land covers.

Table 1 presents the objective results of all studied approaches on the Disko Island dataset. The best indexes are highlighted in bold. In this Table, our method obtains the highest CC value among all methods, which indicates that the fused result of our method is closer to the reference image. For the SAM index, it is found from Table 1 that the SAM value obtained by our method is the smallest. This demonstrates that the spectral curves of different objects between the reference image and the fused result are more similar. For the RMSE index, our method yields the smallest value among all approaches, which illustrates that the difference between the reference image and the fused result of our method is smaller compared to the compared approaches. For the ERGAS index, our method also produces the smallest value, which indicates the overall quality of the fused image obtained by our method is better than other resulting images. Generally, our method can obtain the best fusion performance among all considered approaches in terms of visual result and objective quality. This is due to the fact that our method can accurately estimate the reflectance and illumination components, which makes the proposed method have a stronger ability in injecting the spatial details of the RGB image into HSI.

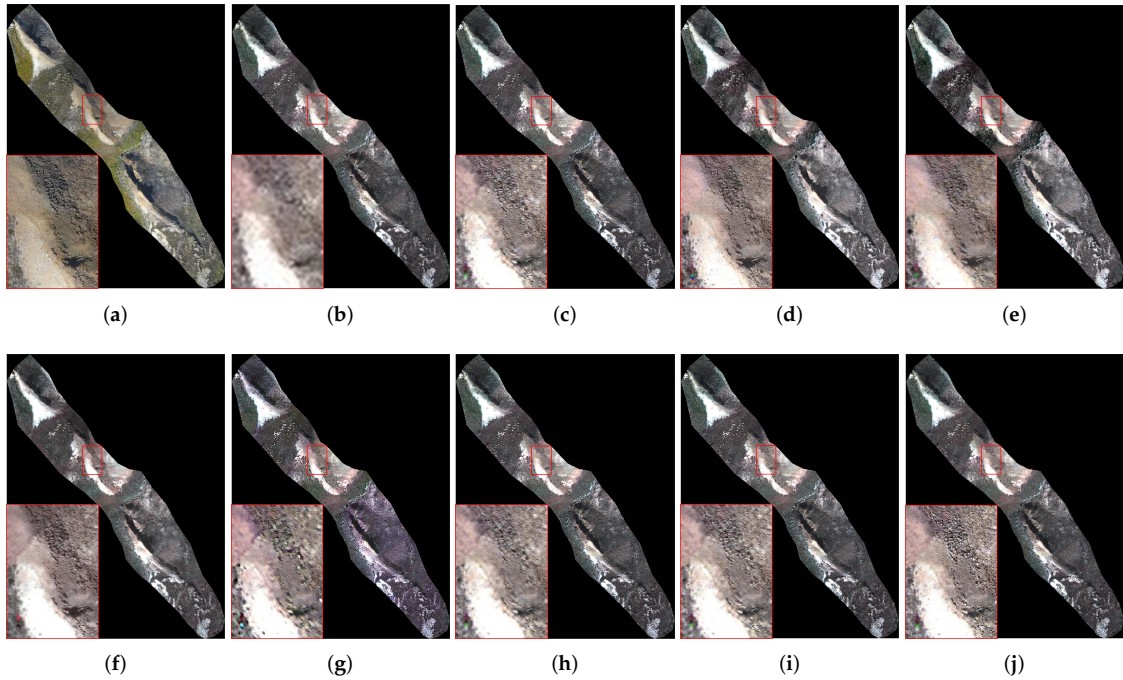

**Figure 6.** Results obtained by different resolution enhancement methods on Disko dataset. (**a**) RGB. (**b**) Three bands of HSI (R:24, G:12, B:6). (**c**) SFIM [25]. (**d**) GS [24]. (**e**) PCA [45]. (**f**) CNMF [36]. (**g**) HySure [48]. (**h**) MTF_GLP [46]. (**i**) MGH [47]. (**j**) Our method.

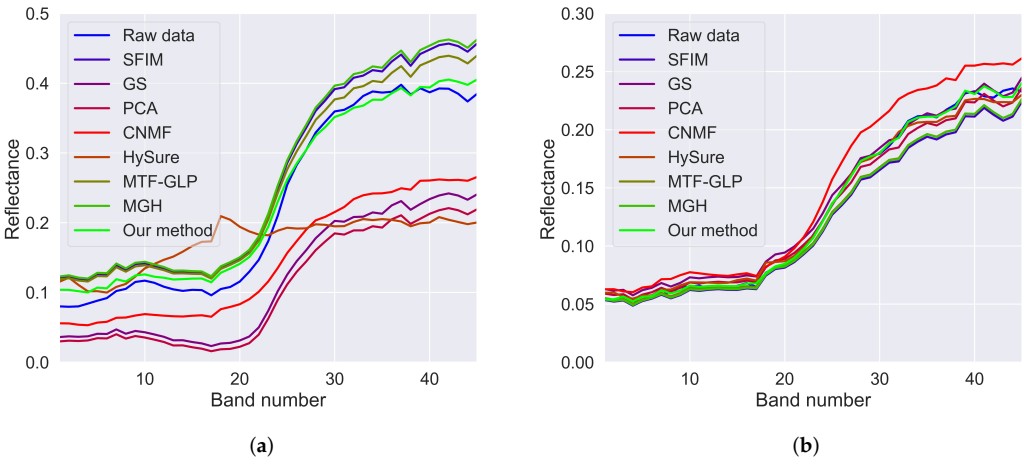

**Figure 7.** Spectral reflectance values of different methods on Disko dataset at two positions. (**a**) Pixel (511, 294). (**b**) Pixel (955, 702).

**Table 1.** Objective quality of the smoothing filter-based intensity modulation (SFIM) [25], Gram-Schmidt (GS) [24], principal component analysis (PCA) [45], coupled nonnegative matrix factorization (CNMF) [36], HySure [48], modulation transfer function-generalized Laplacian pyramid (MTF_GLP) [46], high pass modulation (MGH) [47], and our method on Disko dataset. The best performance is highlighted with bold. The second best performance is highlighted with underscore.

| Indexes | Best | SFIM | GS | PCA | CNMF | HySure | MTF_GLP | MGH | Our Method |
|---------|------|-------|--------|--------|--------|--------|---------|--------|------------|
| CC | 1 | 0.957 | 0.937 | 0.932 | 0.962 | 0.948 | <u>0.963</u> | 0.932 | **0.967** |
| SAM | 0 | <u>1.032</u> | 1.487 | 1.707 | 1.469 | 3.261 | 1.121 | 1.071 | **0.744** |
| RMSE | 0 | 0.029 | 0.034 | 0.035 | 0.023 | 0.031 | <u>0.024</u> | 0.074 | **0.023** |
| ERGAS | 0 | 12.003 | 12.707 | 13.179 | <u>8.358</u> | 11.334 | 8.621 | 37.466 | **8.061** |

### 4.3.2. Litov Dataset

The second experiment is tested on the Litov dataset. Figure 8 shows the resolution enhancement results of different methods. It is found from Figure 8 that the GS and PCA techniques yield unsatisfactory visual results in preserving the spectral information, such as the tree region (see Figure 8d,e). The CNMF method suffers from spectral distortion (see the trees in Figure 8f). The HySure method changes the colors of land covers (see the sand region in Figure 8g). For the MGH method, the edge information of original HSI cannot be well preserved (see Figure 8i). By contrast, the proposed method can better integrate the spatial details of RGB image into HSI (see the local enlarged region in Figure 8j). Furthermore, the spectral reflectance of different methods at two spatial positions is shown in Figure 9. As shown in this figure, the reflectance values of the GS, PCA, and CNMF methods are far from the one of the real HSI. The reflectance curve obtained by our method is very similar to the spectral reflectance of the original HSI, which confirms that our method is able to well retain the spectral information of original HSI with respect to other methods.

To intuitively display the advantage of our method, Figure 10 presents the error images between each fused result and the reference data at 20th band. The less information the error image contains, the better fusion effect this method has. It is easy to observe from Figure 10 that the GS, PCA, and CNMF approaches cannot well merge the spatial details into the HSI (see the local enlarged region in Figure 10b–d). The HySure method introduces artifacts in the edges. The MTF_GLP and MGH methods slightly improve the spatial details. However, the spatial details of the RGB image fail to be merged well into the HSI. By contrast, we can observe from the error images in Figure 10 that the error image obtained by the proposed approach contains the least information, which demonstrates that our approach can effectively merge the RGB and HSI data to obtain high-resolution HSI.

Objective performance of all test approaches on Litov dataset are displayed in Table 2. We can observe that our approach yields the highest CC and the smallest SAM, RMSE, and ERGAS, which also further demonstrates that our method outperforms other studied approaches. These experimental results verify that our approach produces the best fused result in both subjective and objective aspects.

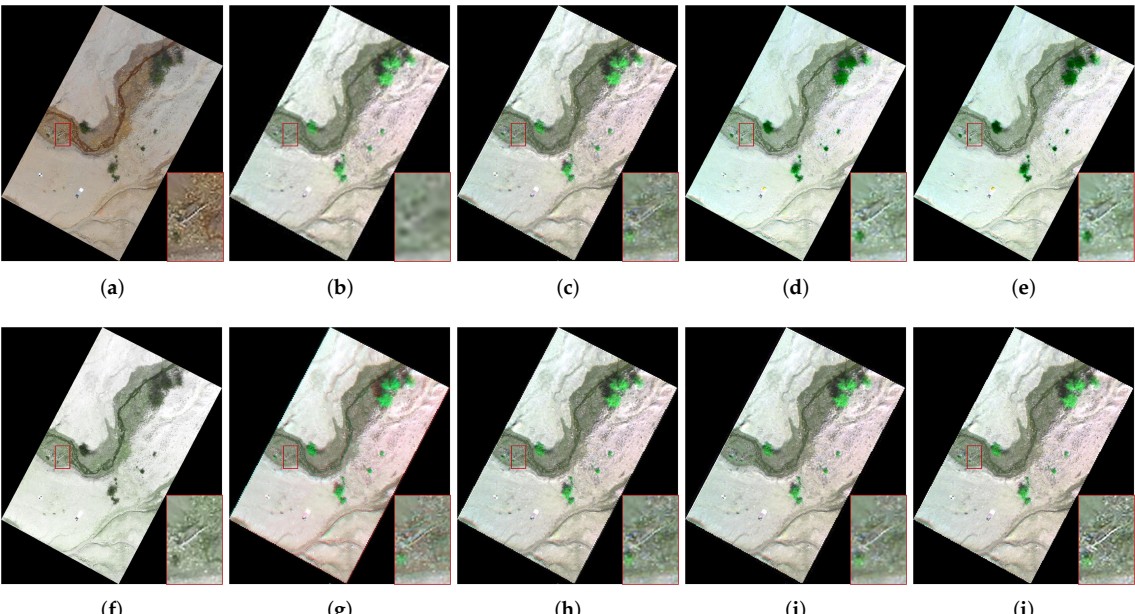

**Figure 8.** Resulting images obtained by different resolution enhancement methods on Litov dataset. (**a**) RGB. (**b**) Three bands of HSI (R:20, G:30, B:10). (**c**) SFIM [25]. (**d**) GS [24]. (**e**) PCA [45]. (**f**) CNMF [36]. (**g**) HySure [48]. (**h**) MTF_GLP [46]. (**i**) MGH [47]. (**j**) Our method.

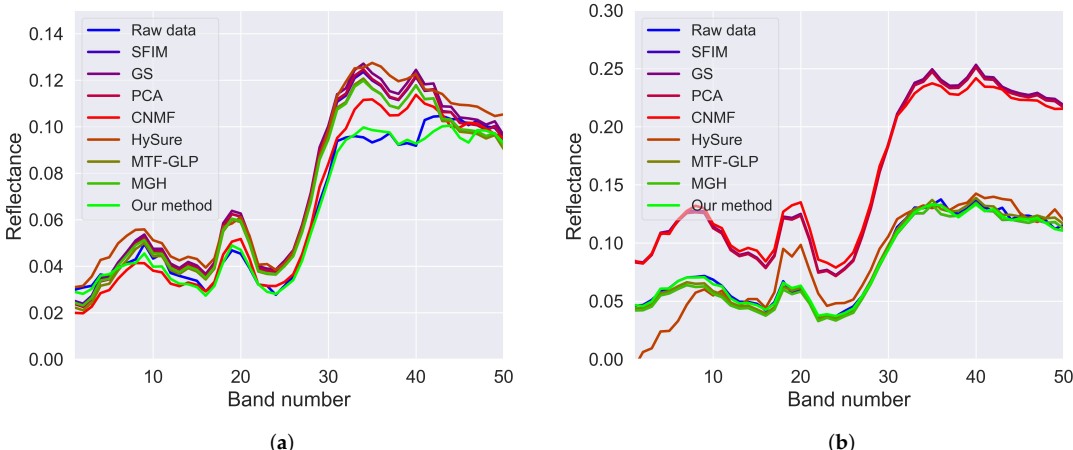

**Figure 9.** Spectral reflectance values of different methods at two positions on Litov dataset. (**a**) Pixel (498, 487). (**b**) Pixel (660, 447).

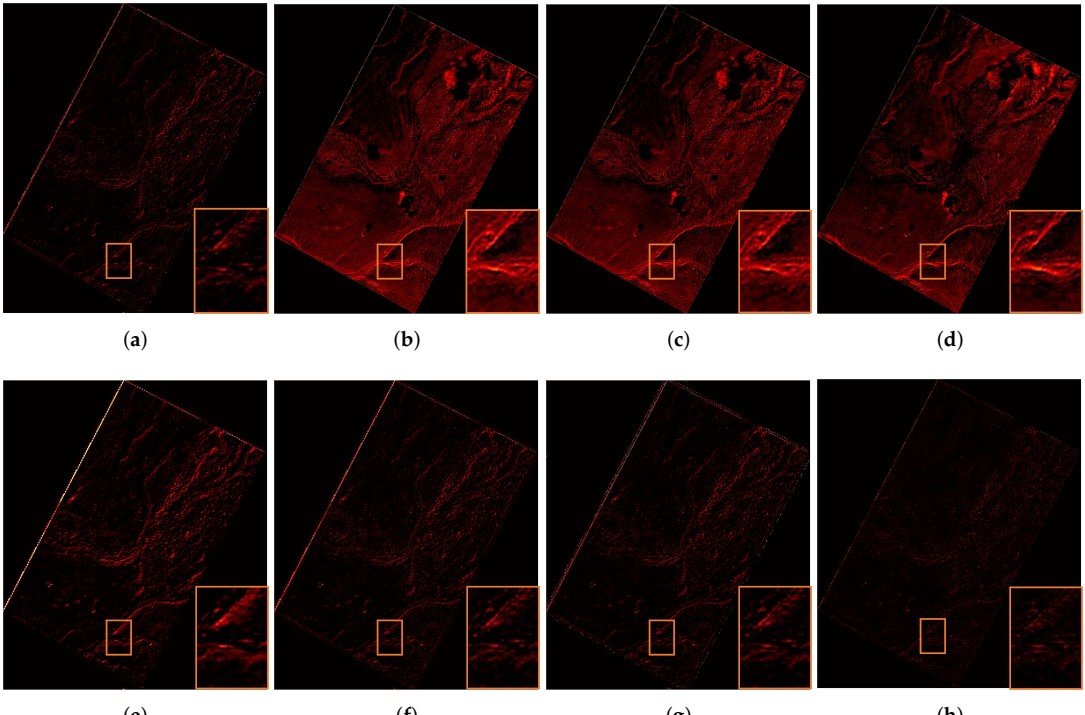

**Figure 10.** The error images between the fused result and the reference image at 20th band using different methods. (**a**) SFIM [25]. (**b**) GS [24]. (**c**) PCA [45]. (**d**) CNMF [36]. (**e**) HySure [48]. (**f**) MTF_GLP [46]. (**g**) MGH [47]. (**h**) Our method.

**Table 2.** Quantitative comparison of the SFIM [25], GS [24], PCA [45], CNMF [36], HySure [48], MTF_GLP [46], MGH [47], and our method on Litov dataset. The best performance is highlighted with bold. The second best performance is highlighted with underscore.

| Indexes | Best | SFIM | GS | PCA | CNMF | HySure | MTF_GLP | MGH | Our Method |
|---------|------|-------|-------|-------|-------|--------|---------|---------|------------|
| CC | 1 | 0.831 | 0.985 | 0.983 | 0.983 | 0.973 | <u>0.986</u> | 0.705 | **0.994** |
| SAM | 0 | 1.982 | 2.332 | 2.422 | 2.031 | 3.376 | <u>1.831</u> | 1.980 | **1.056** |
| RMSE | 0 | 0.175 | 0.017 | 0.018 | 0.019 | 0.023 | <u>0.016</u> | 0.330 | **0.010** |
| ERGAS | 0 | 68.56 | 4.394 | 4.613 | 4.474 | 6.145 | <u>4.201</u> | 115.312 | **2.664** |

## 5. Discussion

### 5.1. Computing Time

The computing efficiency of different approaches on all datasets is presented in Table 3. We did all experiments on a laptop with 8 GB RAM and 2.6 GHz using MATLAB 2014a. From Table 3, it is found that our approach is very fast with respect to other compared approaches since the proposed method only applies several dot multiplication and dot division. Therefore, this method can be directly applied in solving practical engineering tasks.

**Table 3.** The computing time of different methods. Each number denotes the execution time in seconds (s). The best performance is highlighted with bold.

| Datasets | SFIM | GS | PCA | CNMF | HySure | MTF_GLP | MGH | Our Method |
|---|---|---|---|---|---|---|---|---|
| Best value | 0 | 0 | 0 | 0 | 0 | 0 | 0 | 0 |
| Disko | 36.81 | 28.06 | 27.55 | 168.05 | 2068.51 | 54.19 | 44.74 | **6.64** |
| Litov | 14.23 | 13.47 | 13.12 | 75.01 | 883.64 | 15.37 | 14.78 | **4.57** |

### 5.2. Mineral Mapping

In this part, in order to examine the performance of the resolution enhancement approaches for mineral mapping, the support vector machine (SVM) [54–58] is adopted as the spectral classifier, in which the radial basis function kernel is utilized. The number of training and test data is given in Table 4. To quantitatively assess the mineral mapping performance of all studied approaches, three popular quantitative metrics are employed [3,59,60], including overall accuracy (OA), average accuracy (AA), and Kappa coefficient.

The experiment is conducted on Disko HSI. Figure 11 shows the mineral mapping results of all approaches before and after resolution enhancement. Figure 11a,b is the classification maps on the raw data, i.e., RGB and original HSI. Figure 11b–j exhibits the mineral mapping results of different resolution enhancement methods on fused results. As shown in Figure 11, the classification map on the RGB image suffers from obvious noise-like mislabeled pixels due to the lack of rich spectral information. By visually comparing the maps of all studied approaches, the proposed approach yields less misclassification. Furthermore, Table 5 provides the objective classification results. It is clear that the proposed approach yields the highest objective indexes. In addition, the proposed approach has the best classification accuracies for the fourth and fifth classes, which are shown in bold typeface in Table 5. In general, the superior performance of the mineral classification step produced by the proposed approach demonstrates the advantage of the proposed resolution enhancement technique. This important factor is a significant part of practical mineral mapping and relevant missions.

**Table 4.** Numbers of train and test samples.

| Classes | Name | Train | Test |
|---|---|---|---|
| 1 | Vegetation | 55 | 104 |
| 2 | Sandstone | 104 | 62 |
| 3 | Basalt | 72 | 46 |
| 4 | Sulphide | 105 | 157 |
| 5 | Debris | 56 | 68 |
| 6 | Sandstone-basalt | 91 | 40 |
| | Total | 483 | 477 |

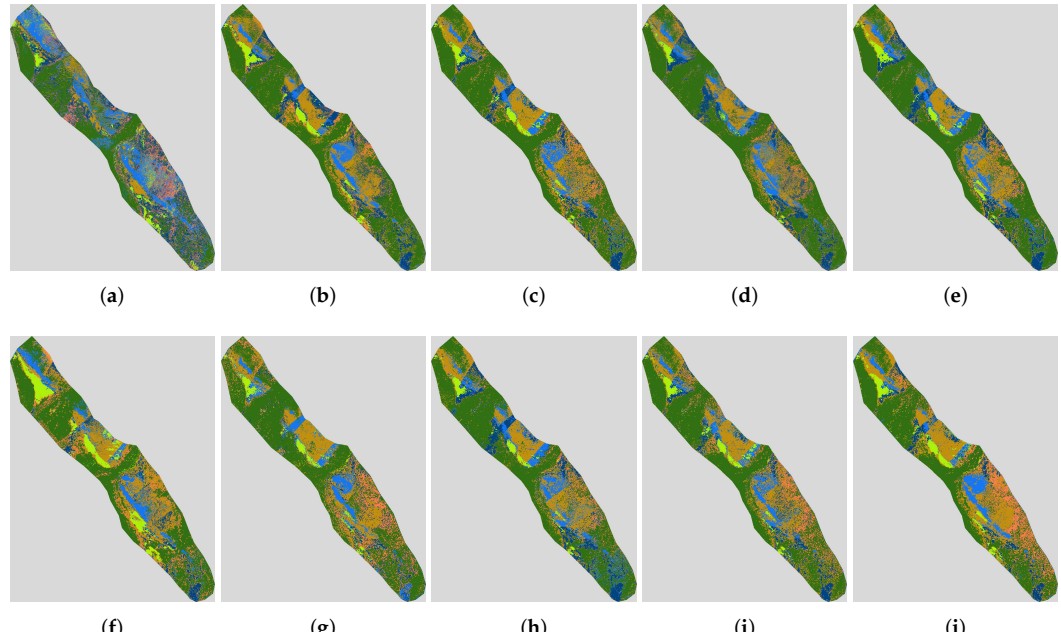

**Figure 11.** Mineral mapping of different approaches with SVM classifier on the Disko dataset. (**a**) RGB. (**b**) HSI. (**c**) SFIM [25]. (**d**) GS [24]. (**e**) PCA [45]. (**f**) CNMF [36]. (**g**) HySure [48]. (**h**) MTF_GLP [46]. (**i**) MGH [47]. (**j**) Our method.

**Table 5.** Classification accuracies on the fused results of different methods, i.e., SFIM [25], GS [24], PCA [45], CNMF [36], HySure [48], MTF_GLP [46], MGH [47], our method and raw data, i.e., RGB and HSI. The best performance is highlighted with bold. The second best performance is highlighted with underscore.

| Class | RGB | HSI | SFIM | GS | PCA | CNMF | HySure | MTF_GLP | MGH | Our Method |
|---|---|---|---|---|---|---|---|---|---|---|
| 1 | 78.79 | 90.00 | 82.35 | **97.12** | 79.49 | 88.18 | 53.76 | <u>87.16</u> | 79.67 | 79.69 |
| 2 | 50.98 | 91.18 | 98.41 | 93.65 | **100.0** | 67.39 | 96.49 | **100.0** | <u>96.88</u> | <u>96.88</u> |
| 3 | 19.54 | 26.04 | <u>32.74</u> | 24.22 | 20.81 | 18.82 | **35.71** | 19.43 | 27.78 | 32.54 |
| 4 | 69.52 | 78.95 | 69.70 | 79.31 | 30.30 | 57.14 | 85.96 | 18.52 | 63.16 | **87.04** |
| 5 | 38.46 | 84.62 | 42.50 | 75.86 | 50.00 | 55.56 | 44.83 | 53.33 | 57.78 | **72.50** |
| 6 | 31.03 | 30.56 | 50.00 | 38.46 | 54.05 | 39.47 | <u>44.44</u> | 54.05 | **58.06** | 55.38 |
| OA | 51.99 | 53.46 | <u>62.47</u> | 58.49 | 52.2 | 49.90 | 57.23 | 52.83 | 61.43 | **66.46** |
| AA | 48.05 | 66.89 | 62.62 | <u>68.10</u> | 55.78 | 54.43 | 60.20 | 55.42 | 63.89 | **70.67** |
| Kappa | 41.38 | 46.23 | <u>55.04</u> | 51.62 | 43.99 | 40.96 | 48.26 | 44.98 | 53.92 | **59.97** |

## 6. Conclusions

In this work, we developed a resolution enhancement method via using the principle of component decomposition to investigate the potential of the hyperspectral data with high resolution for mineral mapping. Based on the principle of the component decomposition, the ideal hyperspectral image is considered as a linear superposition of the reflectance and illumination components. The advantages of the resolution-enhanced HSI were verified by comparing it with other approaches. Objective quality indexes and the visual interpretations prove that the proposed approach can produce a high spectral and spatial resolution, which is helpful for the subsequent mineral mapping step. More importantly, the proposed method is less time-consuming, which can be applied for practical applications. In the future, we will further investigate the potential of multisensor data fusion for mineral mapping.

**Author Contributions:** P.D. is devoted to the methodology, the experiments, and the draft. J.L. provided suggestions and revised the draft. P.G. provided suggestions and carefully modified the manuscript. X.K. carefully revised the manuscript and provided review. J.K. revised the manuscript and analyzed the result. R.J. and R.G. acquired the experimental data, and wrote the data description part. All authors have read and agreed to the published version of the manuscript.

**Funding:** This work is supported by the Major Program of the National Natural Science Foundation of China (No. 61890962), the National Natural Science Foundation of China (No. 61601179), the National Natural Science Fund of China for International Cooperation and Exchanges (No. 61520106001), the Natural Science Foundation of Hunan Province (No. 2019JJ50036), the Fund of Key Laboratory of Visual Perception and Artificial Intelligence of Hunan Province (No. 2018TP1013), and the China Scholarship Council.

**Acknowledgments:** The authors would like to thank the Editor-in-Chief, the anonymous Associate Editor, and the reviewers for their valuable comments and suggestions, which have greatly improved this paper.

**Conflicts of Interest:** The authors declare no conflict of interest.

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
