# Peer review of "Component Decomposition-Based Hyperspectral Resolution Enhancement for Mineral Mapping"

_remotesensing, doi:10.3390/rs12182903_

Round 1

Reviewer 1 Report

The paper describes a new and novel method to fuse high resolution spatial images with lower resolution HSI images.  The paper is well referenced and all details necessary to reproduce the authors' results are included. The approach seems effective and computationally efficient and will find use in the mineral mapping community.

I have the following comments:

Line 26:  why are you comparing and thermal library that acquired data in units of emissivity with that of visible reflectance spectroscopy?

Line 53:  I think you meant joining instead of jointing

Equation 3:  I suppose these are the accepted conversion factors from RGB to YCbCr but you still need a reference.

Line 152:  please provide the wavelength range and spectral width of the bins for the HSI camera.

Line 156:  I assume that the camera was already calibrated radiometrically using a spectrally flat reference, but does this software somehow take into account the influence of the downwelling solar irradiance?  Did you do some other ad hoc calibration using the reflectance of a known material in the image?

Line 158:  I think you mean favourable not favourably

The size of the image analyzed in the Litov data set is not provided.  There is also not a huge difference in image resolution between the RGB and HSI images.

In the first line of Section 3.2 you wrote:  "The reflectance component obtained by intrinsic image composition..."  I think you meant to say "...intrinsic image decomposition..."

Fig. 6:  Providing the number of the spectral band for the composed RGB image using HSI data is not helpful since you never provided the spectral assignments of each channel.  Please provide the wavelengths used for this image.

One last comment:  since the test areas were not large (350 m by 50 m for the Disko data) someone could have walked the imaged area and obtained ground truth which would have been the most convincing argument for the method's efficacy at preserving accurate spectral data.

Reviewer 2 Report

I added my comments in the manuscript attached.

There are many statements for the results in comparison to other methods. Please extend the discussion. The case study results should be further explained and quantified.

Author Response

This manuscript is a resubmission of an earlier submission. The following is a list of the peer review reports and author responses from that submission.